In vitro induction and identification of polyploid Neolamarckia cadamba plants by colchicine treatment

Eng Wee Hiang 1
Ho Wei Seng wsho@unimas.my 1
Ling Kwong Hung 2
1 Faculty of Resource Science and Technology, Universiti Malaysia Sarawak , Kota Samarahan , Sarawak , Malaysia
2 Sarawak Timber Association , Kuching , Sarawak , Malaysia
Balao Francisco
Electronic publication date: 2021 Oct 27
Publication date: 2021
Volume: 9
Electronic Location ID: e12399
Received 2021 Jul 15; Accepted 2021 Oct 6
Copyright: ©2021 Eng et al.
Copyright year: 2021
Copyright holder: Eng et al.
License: This is an open access article distributed under the terms of the Creative Commons Attribution License, which permits unrestricted use, distribution, reproduction and adaptation in any medium and for any purpose provided that it is properly attributed. For attribution, the original author(s), title, publication source (PeerJ) and either DOI or URL of the article must be cited.
License URL: https://creativecommons.org/licenses/by/4.0/

Keywords: Neolamarckia cadamba, Planted forest, In vitro culture, Polyploidization, Colchicine, Polyploid, Anthocephalus cadamba, Flow cytometry, Octoploid, Mixoploid

Funding: The Sarawak Timber Association (STA) GL(F07)/06/2013/STA-UNIMAS(06) GL/F07/STA01/2019 The Ministry of Education Malaysia The funding from the Sarawak Timber Association (STA) was awarded to Wei Seng Ho with grant numbers GL(F07)/06/2013/STA-UNIMAS(06) and GL/F07/STA01/2019. The Ministry of Education Malaysia granted financial assistance to Wee Hiang Eng. The funder has approved the publication of this manuscript.

==============================
Polyploidization has played a crucial role in plant breeding and crop improvement. However, studies on the polyploidization of tropical tree species are still very scarce in this region. This paper described the in vitro induction and identification of polyploid plants of Neolamarckia cadamba by colchicine treatment. N. cadamba belongs to the Rubiaceae family is a natural tetraploid plant with 44 chromosomes (2n = 4x = 44). Nodal segments were treated with colchicine (0.1%, 0.3% and 0.5%) for 24 h and 48 h before transferring to shoot regeneration medium. Flow cytometry (FCM) and chromosome count were employed to determine the ploidy level and chromosome number of the regenerants, respectively. Of 180 colchicine-treated nodal segments, 39, 14 and 22 were tetraploids, mixoploids and octoploids, respectively. The highest percentage of polyploidization (20% octoploids; 6.7% mixoploids) was observed after treated with 0.3% colchicine for 48 h. The DNA content of tetraploid (4C) and octoploid (8C) was 2.59 ± 0.09 pg and 5.35 ± 0.24 pg, respectively. Mixoploid plants are made up of mixed tetraploid and octoploid cells. Chromosome count confirmed that tetraploid cell has 44 chromosomes and colchicine-induced octoploid cell has 88 chromosomes. Both octoploids and mixoploids grew slower than tetraploids under in vitro conditions. Morphological characterizations showed that mixoploid and octoploid leaves had thicker leaf blades, thicker midrib, bigger stomata size, lower stomata density, higher SPAD value and smaller pith layer than tetraploids. This indicates that polyploidization has changed and resulted in traits that are predicted to increase photosynthetic capacity of N. cadamba. These novel polyploid plants could be valuable resources for advanced N. cadamba breeding programs to produce improved clones for planted forest development.

Introduction

Neolamarckia cadamba (Roxb.) Bosser or Anthocephalus cadamba Roxb synonym is a fast-growing multipurpose tree, self-pruning and growing vigorously in exploited areas, such as logged-over forests and plantations (Krisnawati, Kallio & Kanninen, 2011; Ho, Pang & Julaihi, 2014; Nordahlia et al., 2014; Ho, Ling & Pang, 2020). It belongs to the Rubiaceae family and is a natural tetraploid plant with 44 chromosomes (2n = 4x = 44) (Bedi, Bir & Gill, 1981; Kienh & Lorence, 1996; Puangsomlee & Puff, 2001). It is a lightweight hardwood with a density of 290–560 kg/m3 at 15% moisture content. The wood has a moderately fine to medium texture that is easy to work with hand and machine tools to give a perfect surface. This makes it an excellent raw material for plywood production, light construction work and furniture manufacturing (Jøker, 2000; Lim & Chung, 2002; Orwa et al., 2009; Lal et al., 2010). It is also suitable for reforestation and enrichment planting schemes (Krishnapillay, Razak & Appanah, 2007). N. cadamba tree bark has also been used in traditional medicine to treat various illnesses (Mondal et al., 2009). Despite being a commercially important commodity, attempts to improve N. cadamba have been limited to gene mutation study using ethyl methanesulphonate (EMS) (Zayed et al., 2014) and physical mutagens, such as gamma rays (Zanzibar & Danu, 2015). Mutation at ploidy level or polyploidization has yet to be studied.

Colchicine is an inhibitor to polymerization of the tubulin in nucleus that can cause polyploidization in cells. It binds with β-tubulin in the microtubule, resulting in disassociation and destabilization of the microtubule (Lu et al., 2012). Microtubule enables translocation of chromosomes during cell division (Hammond, Cai & Verhey, 2008). The formation of polyploid is due to the failure of chromosome pairs to separate during the anaphase of mitosis as mitosis ceases at metaphase. This will result in two sets of chromosomes in a single cell or the formation of polyploid cells (Inoue, 1952; Eigsti & Dustin, 1956). To date, colchicine is widely used in polyploid induction with different explants, concentrations, treatment durations and methods (Hassan et al., 2020; Julião et al., 2020; Li et al., 2020; Zhang & Gao, 2020). Polyploidization of tropical woody tree species has been rare due to slow-growing habits, long life cycles and lack of in vitro regeneration protocols.

The evolution and formation of new plant species are caused by natural and gradual adaptation to changing environments over a long period of time. Polyploidization increases the number of dominant alleles, diminishing the effects of recessive alleles that often result in undesirable traits (Comai, 2005; Soltis et al., 2015). This hypothesis makes polyploidization an attractive option for plant breeding to produce superior plants. Some induced polyploids were proven to adapt and tolerate better than their progenitors under undesirable or hostile environments. The induced tetraploid of Plumbago auriculata can tolerate cold environment at −5 °C for 24 h (Jiang et al., 2020). In another study, the seed production of Themeda triandra tetraploid was quadrupled and heavier than its diploid under a very dry and hot arid environment (Godfree et al., 2017). The acquired “giga” effects in polyploids increase the size of the cells, followed by doubling in chromosome numbers (Sattler, Carvalho & Clarindo, 2016). This “giga” effect may contribute to greater agronomical gains in polyploid plants by producing larger organs like flowers, fruits, leaves, stems, roots, etc. The fruits of Actinidia chinensis tetraploid increase by 50–60% in size compared with those of the diploid (Wu et al., 2012), while in Thymus vulgaris tetraploid, an increase in the essential oil extracted from the plants was observed (Shmeit et al., 2020). Polyploidization is vital in the creation of triploid plants either within species or between species. Triploid plants are high in demand, especially in the fruit and vegetable industries where seedless fruits are produced (Sattler, Carvalho & Clarindo, 2016; Wang et al., 2016; Hassan et al., 2020). Furthermore, triploids also exhibit heterosis or hybrid vigor (Fu et al., 2014; Urwin, 2014).

Flow cytometry (FCM) analysis is employed in the polyploid identification stage to determine the ploidy level status and DNA content of regenerants treated with colchicine. FCM is an important procedure in polyploidization study where induced polyploids or mixoploids can be segregated from plantlets with unchanged ploidy levels (Bicknell, Boase & Morgan, 2019; Huy et al., 2019; Julião et al., 2020). Mixoploidy exists due to incomplete polyploidization resulting in chimeric tissue comprising a mixture of tetraploid and octoploid cells. The cytometer used in FCM is a high throughput and reliable instrument where a large number of cells can be analyzed in a single run. The FCM, therefore, expedites the detection of representative population and subpopulation of the sample (Doležel, Greilhuber & Suda, 2007; D’Hondt et al., 2011). However, FCM is not efficient in detecting aneuploidy. Therefore, a combination of FCM and karyotype examination is the best way to determine the number of chromosomes and ploidy level. Chromosome count is a straightforward, explicit and standard way of chromosome number determination. Chromosome count can also help to detect the creation of double haploids (Kim et al., 2019), interploids (De Alencar, Azevedo & Latado, 2020) and polyploids (Martin et al., 2019) in plant breeding.

In this study, chromosome doubling or whole-genome duplication of N. cadamba was attempted through in vitro mutagenesis using colchicine. The polyploid plants were identified using FCM and validated using chromosome count followed by morphological characterization. These newly induced polyploid plants may be beneficial for further genetic improvement of N. cadamba to produce improved clones for planted forest development. To the best of our knowledge, this is the first report of polyploidization in N. cadamba.

Materials and Methods

Nodal segments preparation and colchicine treatment

Nodal segments were obtained from shoots regenerated from a selected N. cadamba seedling on regeneration medium (B5 medium + 1.0 mgL−1 BAP + 2% (w/v) sucrose). The nodal segments were treated with colchicine in different concentrations and durations, i.e., 0.1%, 0.3%, and 0.5% of colchicine for 24 h and 48 h, respectively. In the control treatment, sterile distilled water was used for 24 h and 48 h. The nodal segments in the colchicine solution were incubated with shaking using a platform shaker at 100 rpm to improve contact between colchicine and explants. The whole process of colchicine treatment was conducted in the dark at 25 ± 2 °C to avoid deterioration of colchicine under light.

Flow cytometry analysis (FCM) and chromosome count

The colchicine treated nodal segments of N. cadamba were subcultured for five cycles (four weeks per cycle) on shoot regeneration medium (B5 + 1 mgL−1 BAP + 2% (w/v) sucrose) before the ploidy level determination. The fully developed in vitro young leaves from the control and colchicine treated plantlets were collected and used to determine the ploidy level. Untreated plantlets were used as control while Glycine max cv. Polanka was used as an external standard in FCM analysis (Doležel, Binarová & Lucretti, 1989). About 1 cm2 of an in vitro young leaf for each sample was chopped in 1 ml of lysis buffer or LB01 using a scalpel on the disposable petri dish. The LB01 buffer was made up of 80 mM KCl, 20 mM NaCl, 15 mM Tris, 15 mM mercaptoethanol, 2 mM Na2EDTA, 0.5 mM spermidine, 0.1% Triton X-100, pH 7.5, 50 µg propidium iodide and 50 µg RNase (Doležel, Binarová & Lucretti, 1989). The finely chopped leaf pieces of each sample were filtered through a 50-µm cell strainer separately into a 5 mL flow cytometry tube. The filtrate of each sample was analyzed by a FACSCalibur flow cytometer - Becton Dickinson equipped with an argon-ion laser (15 mW) at 488 nm. Histograms for each sample were collected over 1,024 channels. DNA content of each sample can be estimated using this formula (Doležel, Greilhuber & Suda, 2007): SPL = (S(G0/G1)/R(G0/G1)) × RPL, where SPL is 2C value of colchicine treated sample; RPL is 2C of the control sample, i.e., Glycine max Merr. ‘Polanka’ with a value 2.50; S(G0/G1) means the position of colchicine treated sample, and R(G0/G1) means the position of control sample peak.

The 5-day-old in vitro root tips after transplantation were excised at 1 cm in length from the tip of the roots. The root tips were washed with running tap water to eliminate the agar medium. In pre-treatment, these root tips were first submerged in 0.002 M 8-hydroxyquionline for 2 h at 4 °C followed by another 2 h at room temperature. The pre-treated root tips were washed three times with distilled water. In fixation, the root tips were dipped in freshly prepared cold Carnoy’s solution 1 (Ethanol 95%: glacial Acetic Acid; 3: 1) for 24 h at 4 °C. After fixation, the root tips were washed with distilled water three times. The hydrolysis of the pre-treated and fixed root tips took place using 1 N HCl for 15 min at 60 °C. After HCl hydrolysis, the root tips were washed with distilled water three times. The staining process was carried out at room temperature. Feulgen stain was applied on the root tips for 30 min in dark condition before squashing with a metal bar in a drop of aceto-orcein 1% stain followed by thumb squash (Eng, Ho & Ling, 2020).

In vitro propagation and acclimatization

In vitro nodal segments were excised from N. cadamba tetraploid, mixoploid and octoploid plantlets and cultured on shoot regeneration medium (B5 + 0.8 mgL−1 BAP + 2% (w/v) sucrose) for six weeks with subculture once every three weeks. The number of shoots produced and the percentage of nodal segments producing shoots were recorded. After regeneration, well-developed shoots, about 1 cm in height, of the tetraploid, mixoploid and octoploid cultures were excised from the shoot clumps and transferred to basal B5+ 2% (w/v) sucrose medium for eight weeks with subculture once every four weeks. Morphological characters of tetraploid, mixoploid and octoploid shoots were recorded, namely plant height, number of leaves, number of nodes and stem diameter after eight weeks of culture. The shoots of the tetraploid, mixoploid and octoploid cultures with the height of 3.0 cm were excised and cultured on rooting medium, 12 B5 + 0.1 mgL−1PBZ (Paclobutrazol) + 2% (w/v) sucrose. The number of roots and the percentage of rooting were recorded after two weeks of culture. Well rooted in vitro plantlets were cleaned thoroughly under running tap water before planted onto different acclimatization media. These acclimatization media are Jiffy-7, loamy soil and soil mixture (loamy soil: compost = 3:1). The Jiffy-7 pellets were soaked in tap water overnight before use. Both the loamy soil and the soil mixture were sterilized by autoclaving at 121 °C for 20 min. All transferred plantlets were kept in air-tight containers for the first week before the container’s covers were removed gradually to regulate humidity and avoid plantlets from wilting. Plantlets were maintained under room temperature. Fungicide was applied weekly to prevent fungal contamination among the acclimatizing plantlets. The number of surviving plantlets in different media was recorded after four weeks of acclimatization.

This experiment consisted of tetraploid, mixoploid and octoploid plantlets; each genotype consisted of six plantlets, and the experiment was repeated thrice. The data collected were subjected to ANOVA using the IBM SPSS version 23 program. The ANOVA was carried out first, followed by Duncan New Multiple Range Test (DNMRT) (p < 0.05).

Morphological characteristics and soil plant analysis development (SPAD) value

The well-developed third leaf from the shoot tip of one-month-old acclimatized seedlings was obtained for leaf and midrib thickness studies. Using a razor blade, cut across the width of the leaf sample about 1 mm wide. The leaf section was placed on a slide vertically with the cut section facing upwards. The leaf section was observed using a LEICA DM500 compound light microscope 4× objective lens. The image of the leaf section was captured using a LEICA ICC50 HD camera. LEICA Application Suite (LAS EZ) was used to measure the leaf and midrib thickness. A total of 10 leaves for each genotype was analyzed, namely tetraploid, mixoploid and octoploid to determine leaf and midrib thickness.

The well-developed leaves of acclimatized seedlings were obtained for abaxial leaf study. A layer of clear nail polish was painted on the abaxial leaf side and leave it to dry. Then a pair of tweezers was used to peel the nail polish from the leaf surface. Place the nail polish peel with leaf impression on a slide. The leaf surface was observed using a LEICA DM500 compound light microscope 40× objective lens, and the image was captured using a LEICA ICC50 HD camera. A total of 100 stomata was measured for each genotype, namely tetraploid, mixoploid and octoploid. LEICA Application Suite (LAS EZ) was used to measure the length and width of the stomata. Ten optical micrographs (0.33 × 0.25 mm) were used to determine the density of the stomata and trichomes. ImageJ software was used to determine the number of stomata and trichomes per mm2.

Scanning electron microscope (SEM) was used to examine the structure and other details of the stoma and trichomes. Initially, six-month-old leaves were sectioned using a blade, and only the middle part of the leaf sections was used for the SEM study. Leaf samples were prepared using the protocol developed by Talbot & White (2013). The leaf sections were fixed in 100% methanol for 10 min. After the fixation, they were dehydrated two times in 100% methanol for 30 min before further dried using a critical point drying machine. The leaf sections were mounted on SEM stud with the abaxial part facing upwards. The samples were then coated with gold coatings before they were observed using SEM by JEOL model JSM-6390OLA.

A mature stem from a one-year-old plant was cut using a fine tooth saw to study the cross-section of the 10-month-old stem. The thickness of bark, pith and wood of N. cadamba were recorded with the aid of a pair of vernier calliper. A total of three stem sections for each genotype were analysed, namely tetraploid, mixoploid and octoploid.

The leaf SPAD values were measured using portable chlorophyll meter, model TYS-B (Mindfull Technology Co. Ltd) that was equipped with two light sources, namely red light (650 nm) and infrared light (940 nm). To measure SPAD value, grip the leaf sample between the pressure head and measuring head of the meter. The buzzer of this meter will buzz for less than 3 s signalling the measurement is completed and the reading can be recorded. A total of 30 trees consisting of tetraploid, mixoploid and octoploid plants were measured, and each genotype consisted of 10 plants.

The data collected were subjected to ANOVA using the IBM SPSS version 23 program. The ANOVA was carried out first, followed by Duncan New Multiple Range Test (DNMRT) (p < 0.05).

Results

Flow cytometry analysis (FCM) and chromosome count

In all colchicine treatments (0.1%, 0.3% and 0.5%), the longer treatment duration at 48 h tend to produce a higher number of octoploids and a lower number of mixoploids (Table 1). In total, 39 tetraploids, 14 mixoploids and 22 octoploids were produced in this study. Two treatments, namely 0.3% colchicine for 48 h and 0.5% colchicine for 48 h gave higher percentages (20%) of octoploids. However, treatment with 0.3% colchicine for 48 h produced the best result that yielded 20.0% octoploids and 6.7% mixoploids. In the 0.3% colchicine treatment for 24 h, the highest percentage of mixoploids was obtained. Generally, a combination of a higher concentration of colchicine and a longer treatment duration yielded a higher percentage of octoploids in N. cadamba. The ploidy level of regenerants of N. cadamba can be determined based on the histogram generated using a flow cytometer (Figs. 1A–1C). The histogram showed a single dominant peak in control and tetraploid plants with the mean position of a peak for 4x (Fig. 1A). For the mixoploid plants, the histogram showed two separate dominant peaks with the mean position of the first peak for 4x and the mean position of the second peak for 8x (Fig. 1B). For the octoploid plants, the histogram showed a single dominant peak with the mean position of a peak for 8x (Fig. 1C). It was also noted that the octoploid cells (Fig. 1C) are two times greater in the mean position of the peak than those of the tetraploid cells (Fig. 1A). Our data showed that the estimated DNA content for tetraploid (4C DNA) is 2.59 ± 0.09 pg, while for octoploids, there is a two-fold increase in the estimated DNA content (8C DNA = 5.35 ± 0.24 pg).

Table 1 Polyploidization rate of nodal segments of N. cadamba using different colchicine concentrations and treatment durations.

Colchicine (%, w/v)	Treatment duration (h)	Number of explants	Number of surviving explants	Number of tetraploids (%)	Number of mixoploids (%)	Number of octoploids (%)	
0.0	24	30	27 (90.0)	27 (90.0)	0 (0.0)	0 (0.0)	
	48	30	20 (66.7)	20 (66.7)	0 (0.0)	0 (0.0)	
0.1	24	30	18 (60.0)	16 (53.3)	2 (6.7)	0 (0.0)	
	48	30	16 (53.3)	11 (36.7)	2 (6.7)	3 (10.0)	
0.3	24	30	14 (46.7)	6 (20.0)	6 (20.0)	2 (6.7)	
	48	30	12 (40.0)	4 (13.3)	2 (6.7)	6 (20.0)	
0.5	24	30	8 (26.7)	2 (6.7)	1 (3.3)	5 (16.7)	
	48	30	7 (23.3)	0 (0.0)	1 (3.3)	6 (20.0)	
Notes.

Values in parentheses represent the percentage of corresponding figures.

Figure 1 Polyploid plants detection using FCM and chromosome count of N. cadamba.

(A) FCM histogram for control or tetraploid plantlet (2n = 4x), (B) FCM histogram for mixoploid plantlet (2n + 4n = 4x + 8x), (C) FCM histogram for octoploid plantlet (4n = 8x), (D) Tetraploid cell with 44 chromosomes (Bar = 5 µm) and (E) Octoploid cell of N. cadamba “Kestamas-19” with 88 chromosomes (Bar = 5 µm).

The ploidy level of samples determined using FCM was verified using the optimized chromosome count procedure. The micrograph of the somatic cells of N. cadamba root tips showed that both the tetraploid and the control contain 44 chromosomes per nucleus (Fig. 1D); the octoploid (a.k.a N. cadamba “Kestamas-19”) contains 88 chromosomes per nucleus (Fig. 1E), while the mixoploid contains either 44 or 88 chromosomes per nucleus. The ploidy level results determined using the FCM procedure agree with results obtained using the optimized chromosome count method.

In vitro propagation and acclimatization

The shoot regeneration ability between the tetraploid, mixoploid and octoploid was observed and compared with each other (Table 2). The mean number of shoots produced was significantly higher in the tetraploid (5.4) as compared to mixoploid (4.3) and octoploid (4.2) (Table 2). In terms of the mean percentage of explants producing shoots, the tetraploid (94.4%) and mixoploid (100.0%) showed significantly higher mean percentages of explants producing shoots compared to the octoploid (83.3%). The new genotypes of N. cadamba, i.e., the mixoploid (Fig. 2B) and octoploid (Fig. 2C) showed significantly lower achievement in terms of the number of shoots produced as compared with the tetraploid (Fig. 2A). The octoploid responded with the lowest mean percentage of regeneration among the three genotypes (Table 2). The observation from this study concluded that polyploidization resulted in higher mortality in the shoot regeneration medium.

Table 2 In vitro shoot regeneration, growth, rooting and acclimatization of tetraploids, mixoploids and octoploids of N. cadamba.

Morphological characteristics	Tetraploid	Mixoploid	Octoploid	
Regeneration				
Number of shoots ± SE	5.4 ± 0.1b	4.3 ± 0.2a	4.2 ± 0.3a	
Regeneration (%) ± SE	94.4 ± 5.6b	100.0 ± 0.0b	83.3 ± 0.0a	
Growth				
Stem height (cm) ± SE	7.2 ± 0.0c	6.8 ± 0.0b	4.4 ± 0.4a	
Number of leaves ± SE	12.8 ± 0.2a	12.9 ± 0.2a	13.0 ± 0.3a	
Number of nodes ± SE	6.4 ± 0.1a	6.4 ± 0.1a	6.5 ± 0.0a	
Stem diameter (mm) ± SE	2.3 ± 0.1a	2.7 ± 0.1b	2.9 ± 0.1b	
Rooting				
Root length (cm) ± SE	1.8 ± 0.1c	1.6 ± 0.0b	1.4 ± 0.0a	
Number of roots ± SE	13.8 ± 0.9b	16.1 ± 0.5b	11.1 ± 0.7a	
Rooting (%) ± SE	100.0 ± 0.0a	100.0 ± 0.0a	100.0 ± 0.0a	
Acclimatization percentage				
Jiffy-7 ± SE	94.4 ± 5.6b	94.4 ± 5.6b	94.4 ± 5.6b	
Loamy soil ± SE	88.9 ± 5.6b	94.4 ± 5.6b	88.9 ± 5.6b	
Loamy soil: compost 1:3 ± SE	50.0 ± 9.6a	61.1 ± 5.6a	66.7 ± 0.0a	
Notes.

For each parameter, means with different letters indicate significantly different at 5% level of probability using Duncan’s test (p < 0.05).

Figure 2 In vitro culture and acclimatization of N. cadamba polyploid plants.

Shoot regeneration of tetraploids (A), mixoploids (B) and octoploids (C). (D) Shoot tip growth of tetraploids (left), mixoploids (middle) and octoploids (right). Rooting of tetraploids (E), mixoploids (F) and octoploids (G). (H) Acclimatization of plantlets (T = Tetraploid, M = Mixoploid, O = Octoploid) on different media (J = Jiffy-7, S = Soil, C = Soil: Sompost 3: 1), (Bar = 1 cm).

The growth of N. cadamba tetraploid, mixoploid and octoploid shoot tips on the basal B5 medium was evaluated after they have been transferred from the regeneration medium. After eight weeks of growth on the basal B5 medium, the tetraploid attained the highest mean plant height at 7.2 cm, followed by the mixoploid at 6.8 cm and the octoploid at 4.4 cm only (Table 2). Polyploidization in N. cadamba has noticeably reduced the plant height (Table 2 and Fig. 2D). There was no significant difference between the tetraploid, mixoploid, and octoploid shoots in terms of the mean number of leaves and the mean number of nodes. The mean number of leaves ranges from 12.8 to 13.0, while the mean number of nodes ranges from 6.4 to 6.5 for tetraploid, mixoploid and octoploid (Table 2). The octoploid plantlets have shorter stems making each of them an interestingly compact architecture (Fig. 2D). The mean stem thickness of the shoots in tetraploid was 2.3 mm only, while the octoploid 2.9 cm and mixoploid 2.7 mm. The stem of the two new polyploids was significantly thicker than the tetraploid (Table 2). From the in vitro growth parameters recorded, polyploidization resulted in the reduction of stem height but has the opposite effect of increment in stem diameter. The octoploids are sturdier than mixoploid and tetraploid plantlets due to their shorter stems.

The excised shoots of tetraploid, mixoploid and octoploid of N. cadamba were cultured in rooting medium for 14 days to compare the rooting capability of plantlets of different ploidy levels (Table 2 and Figs. 2E–2G). All shoots, regardless of ploidy level, produced roots after the 14 days on rooting medium. The tetraploid (13.8) and mixoploid (16.1) produced a higher mean number of roots compared with the octoploid (11.1). In terms of mean root length, the tetraploid produced the longest mean root length (1.8 cm), followed by the mixoploid (1.6 cm) and octoploid (1.4 cm) (Table 2). In this study, the shoots were cultured on the rooting medium for no longer than 14 days to avoid excessive rooting, leading to a high number of roots and longer root length.

The rooted plantlets were transferred to three different media types, namely Jiffy-7, loamy soil, soil mix (loamy soil: compost; 3: 1) to assess the percentage of acclimatization of the plantlets (Fig. 2H). The mean percentage of acclimatization of the tetraploid plantlets planted on Jiffy-7 pellets and loamy soil was 94.4% and 88.9%, respectively. These two media types were significantly superior to the soil mix that produced only 50.0% mean percentage of acclimatization. A similar trend was observed in the mixoploid and octoploid plantlets (Table 2). Two factors that caused mortality to the N. cadamba plantlets during acclimatization were water stress and fungal contamination. Water stress can be controlled by placing plantlets in an enclosed container to maintain higher humidity and prevent water loss. In this study, 5 gL−1 Benocide solutions were sprayed with a mist sprayer. Incorporating compost in the soil mix should be avoided in this experiment as it would encourage fungal contamination even after autoclaving before planting.

Morphological characteristics and SPAD value

The N. cadamba octoploid (247.2 µm) had thicker leaves than its tetraploid counterpart (160.8 µm). However, there was no significant difference in mean leaf thickness between the mixoploid and tetraploid. The mean midrib vertical thickness of the octoploid (831.7 µm) was higher than both the mixoploid (762.4 µm) and tetraploid (666.6 µm). The mean midrib horizontal thickness of the octoploid (899.1 µm) and the mixoploid (793.0 µm) was higher than the tetraploid (616.2 µm). However, there was no significant difference between the mixoploid and the tetraploid in terms of mean midrib vertical thickness (Table 3 and Figs. 3A–3C).

Table 3 Morphological characteristics and SPAD values.

Characteristics	Tetraploid	Mixoploid	Octoploid	
Leaf thickness				
Leaf thickness (µm) ± SE	160.8 ± 38.5a	197.0 ± 8.7ab	247.2 ± 15.0b	
Midrib vertical thickness (µm) ± SE	666.6 ± 25.0a	762.4 ± 38.1ab	831.7 ± 61.7c	
Midrib horizontal thickness (µm) ± SE	616.2 ± 42.1a	793.0 ± 48.5b	899.1 ± 56.0b	
Stomata and trichome				
Stomata density (mm−2) ± SE	426.7 ± 9.4b	298.2 ± 6.3a	301.8 ± 10.9a	
Trichome density (mm−2) ± SE	33.9 ± 4.0b	4.9 ± 2.0a	4.9 ± 2.7a	
Stomata length (µm) ± SE	25.3 ± 0.2a	29.3 ± 0.3b	30.7 ± 0.3c	
Stomata width (µm) ± SE	14.9 ± 0.2a	16.1 ± 0.2b	16.6 ± 0.2b	
Stem cross-section				
Bark (mm) ± SE	0.4 ± 0.1a	0.4 ± 0.1a	0.3 ± 0.0a	
Wood (mm) ± SE	3.1 ± 0.3a	3.7 ± 0.3a	3.8 ± 0.3a	
Pith (mm) ± SE	5.7 ± 0.3b	5.2 ± 0.1b	3.6 ± 0.2a	
SPAD value				
SPAD ± SE	37.1 ± 1.0a	37.2 ± 0.4a	39.7 ± 0.6b	
Notes.

For each parameter, means with different letters indicate significantly different at 5% level of probability using Duncan’s test (p < 0.05).

Figure 3 Morphological characteristics of N. cadamba polyploid plants.

Leaf sections of tetraploids (A), mixoploids (B) and octoploids (C). Stomata imprints of tetraploids (D), mixoploids (E) and octoploids (F). Stoma micrographs of tetraploids (G), mixoploids (H) and octoploids (I) under SEM. Trichome micrographs of tetraploids (J), mixoploids (K) and octoploids (L) under SEM. (M) Stem cross-section of tetraploids (left), mixoploids (middle) and octoploids (right) (b = bark; w = wood; p = pith). (T = Leaf thickness, V = Midrib vertical thickness, H = Midrib horizontal thickness) (Bar A–C = 0.5 mm) (Bar D–F = 0.05 mm).

The mean stomata density per mm−2 in octoploid (301.8) and mixoploid (298.2) were significantly less than the tetraploid (426.7) of N. cadamba plants (Table 3 and Figs. 3D–3F). Similarly, the mean trichome density mm−2 was significantly less in octoploid (4.9) and mixoploid (4.9) when compared with the tetraploid (33.9) of N. cadamba plants. The mean stomata length of either the mixoploid (29.3 µm) or octoploid (30.7 µm) was higher than the tetraploid (25.3 µm). The mean stomata width of either the mixoploid (16.1 µm) or octoploid (16.6 µm) was also higher than the tetraploid (14.9 µm). Generally, the mixoploid and octoploid plants of N. cadamba possess a bigger stomata size than the tetraploid plant (Table 3 and Figs. 3G–3I). The stoma of N. cadamba tetraploid, mixoploid and octoploid was then observed using SEM at 2000x. The micrograph showed that regardless of ploidy level, the stoma is made up of two complementary kidney-shaped guard cells arranged in opposite sites to form an aperture. Each guard cell is flanked by a subsidiary cell that aligns parallel to its long axis. The trichomes of N. cadamba in this study were observed using SEM at 2000x. The micrograph showed, regardless of ploidy level, the trichomes were categorized as unicellular and non-glandular trichomes (Figs. 3J–3L) based on trichomes anatomical categorization by Werker (2000) study. The shape and other attributes of N. cadamba trichome are categorized as ornithorhynchous based on the nomenclatures determined by Payne (1978).

In the stem cross-section study, the bark, wood and pith layers of N. cadamba plants were studied. There was no significant difference between the tetraploid, mixoploid and octoploid in the bark and wood layers. However, the tetraploid (5.7 mm) and mixoploid (5.2 cm) have a bigger pith than the octoploid (3.6 mm) (Table 3 and Fig. 3M).

The mean SPAD value of the octoploid was the highest (39.7) when compared to the mixoploid (37.2) and tetraploid (37.1). Hence, the chlorophyll content of the octoploid was also higher than both the mixoploid and tetraploid (Table 3).

Discussion

Flow cytometry analysis (FCM) and chromosome count

Colchicine is a commonly used chromosome doubling agent. Its anti-mitotic properties result in the disruption of microtubules to produce polyploids. Polyploids induced by using colchicine are also termed as colchiploids. This term was widely used in 1960’s (Sen & Marimuthu, 1960; Raghuvanshi & Joshi, 1964; Das, Prasad & Sikdar, 1970). The optimum amount of colchicine used in polyploid production is wide-ranging, with concentrations ranging from 0.01% (Thao et al., 2003) to 1.0% (Demtsu et al., 2013). The difference is one hundred times between the highest and the lowest colchicine concentration. The percentage of successful polyploidization also varies. The success rate could be attributed to the species under investigation, the colchicine application protocol (in vitro or ex vitro) and the explant used.

FCM can detect the existence of mixoploids and it can analyze a huge population of cells in a relatively short duration. In the histogram generated from FCM for N. cadamba, Fig. 1B, two distinct populations of cells were detected, namely 4x and 8x, and this finding can be interpreted from the two dominant peaks in the histogram. Polyploidization studies conducted without the FCM procedure often yielded very high polyploidization percentage but no occurrence of mixoploids, such as 100% for Pogostemon cablin leaves (Widoretno, 2016), 42.3% for Gossypium arboretum seeds (Yang et al., 2015) and 41.7% for Vitis sp. (Sinski et al., 2014) shoot tips. These studies determined polyploidization through manual chromosome counting, which is difficult because the researcher has to analyze a huge number of cells needed to represent an accurate cell composition of the plant. Studies on polyploidization using seeds, seedlings, shoot tips, and nodal segments are predominantly prone to the production of mixoploids which is an unwelcome result (Eng & Ho, 2019). Studies using these sources of explants should adopt FCM to avoid mixoploids being pooled into induced polyploids or the original ploidy category. FCM will help the researcher to identify the mixoploids. The existence of mixoploids will often mislead the analysis leading to a possible inaccurate conclusion.

According to Ohri, Bhargava & Chatterjee (2004), the estimated 4C DNA for A. cadamba was 2.77 ± 0.27 pg using M86 Vickers microdensitometer. The data have been published and archived in the C-value Kew Garden database (https://cvalues.science.kew.org/), where the data are conveyed in estimated 1C DNA, that is 0.69 pg. These data showed that N. cadamba is a tetraploid. Based on these data, our estimated 4C DNA of N. cadamba was 2.59 ± 0.09 pg, which falls within the published data range, while the octoploids estimated 8C DNA was 5.35 ± 0.24 pg. This study indicated that the DNA content had doubled in N. cadamba tetraploid to form N. cadamba octoploid. There are many uses of estimated DNA content apart from the indication of ploidy level change. According to Bennett, Bhandol & Leitch (2000), estimated DNA content could be used in the molecular investigation, evolution, variation, constancy, phylogenetic, phenotypic, phenological, ecological, environmental indicators, and paleobiological trends.

According to Regalado et al. (2017), mixoploid plants are unstable as competition occurs between the original cells and the polyploid cells leading to the elimination of the latter. As a result, mixoploids status will be reversed to its original ploidy level. In citrus breeding, mixoploids were discarded due to stability concern and low fertility if crossed with other species (Grosser, Kainth & Dutt, 2014). For this study, N. cadamba mixoploids produced were maintained and further evaluated together with tetraploids and octoploids. Mixoploids are often regarded as a failure or undesirable by-products of polyploidization studies so that efforts are made to eliminate them through deliberate means, namely mechanical isolation of putative polyploids (Aleza et al., 2009; Regalado et al., 2015); shoot regeneration using nodal segments (Zhou, Zeng & Yan, 2017) and repeated subcultures of apical buds (Jiang et al., 2020).

The chromosome count enables the verification of samples determined using FCM. In the present study, the FCM results using leaf samples aligned with our findings through the chromosome count using root tip samples. The control, which is a tetraploid (2n = 4x) contains 44 chromosomes per nucleus, the octoploid (4n = 8x) contains 88 chromosomes per nucleus and the mixoploid (2n + 4n = 4x + 8x) contains either 44 or 88 chromosomes per nucleus. This study, based on the chromosome count, has verified the ploidy level data obtained from FCM. The polyploidization of N. cadamba using colchicine has successfully induced octoploids and mixoploids from tetraploid explants. The chromosome number of N. cadamba (2n = 4x) has doubled (4n = 8x) after colchicine treatment (Figs. 1D and 1E). This is a two-fold increment of chromosome number from the original N. cadamba explant. This is the first establishment of N. adamba octoploid ever reported, where the chromosome number is 88.

In vitro propagation and acclimatization

In vitro technique was used in this study to examine the effects of N. cadamba polyploid plants at different stages of micropropagation following the established tissue culture protocol of N. cadamba (Mok & Ho, 2019). With these advantages, mass phenotyping and characterizing of new plants are now a possible task. Our study is in agreement with that of the induced polyploid plants of Cichorium intybus (Ravandi, Rezanejad & Dehghan, 2014) and Dendrobium officinale (Pham et al., 2019). The explants of induced polyploid of D. officinal produced a higher protocorm induction rate but required a longer time for protocorm initiation. According to Ravandi, Rezanejad & Dehghan, (2014), the ability to regenerate shoots from callus or organogenesis in induced polyploid explants was significantly reduced. Tsukaya (2008) reported that Arabidopsis thaliana with more than eight sets of homologous chromosomes will display high ploidy syndrome, exhibiting contrasting effects by producing bigger cell in volume but smaller leaves. Several recent studies have employed the in vitro system to study the effects of plant ploidy level on morphological traits (Javadian et al., 2017; Pham et al., 2019; Shmeit et al., 2020). The present study is in agreement with Pham et al. (2019), where colchicine-induced tetraploids of D. officinale showed shorter stem length and thicker stem. The thicker stem was also found in the colchicine-induced tetraploid of Linum album (Javadian et al., 2017). However, oryzalin-induced tetraploid of Thymus vulgaris showed longer stem length and thicker stem (Shmeit et al., 2020).

The induced octoploid of N. cadamba exhibited slower root formation. The root growth was also slower in the rooting medium (Table 2 and Figs. 2E–2G). According to Mok & Ho (2019), the rooting medium 1/2 B5 + 0.1 mgL−1 PBZ is an optimum medium in inducing a sufficient number of roots in N. cadamba. The PBZ concentration does not affect growth during the rooting, and there is no callus formation on the shoot. The protocol has improved the success rate of acclimatization. PBZ derived from an aromatic compound, trizol is a plant growth regulator with many uses in agriculture. Among others, it is used to induce flowering, fruiting and prevent fungal infections (Wang, Chang-Yi Wu & Lonameo, 2019). PBZ has many uses in in vitro culture for different plants, such as shoot multiplication medium MS + 5 mgL−1 in Phoenix dactylifera (Awadh, Abdulhussein & Almusawi, 2019), rooting medium 1/2 B5 + 0.1 mgL−1 in N. cadamba (Mok & Ho, 2019) and rooting medium MS + 1.5 mgL−1 in Zygopetalum crinitum (Gimenes et al., 2018).

Acclimatization is the final stage of in vitro system that will determine the production number of the plantlets. In vitro acclimatization depends on the capability of the plantlets to transform from being either mixotrophic or heterotrophic to photoautotrophic before they can adapt to the harsh ex vitro conditions. Plantlets in in vitro system are mixotrophic or heterotrophic where complete nutrients are provided to sustain growth, while plantlets under ex vitro conditions need to become photoautotrophic where foods can be synthesized through photosynthesis (Debergh, 1991; Pospíšilová et al., 1999; Chandra et al., 2010). With this principle in mind, in vitro plantlets of N. cadamba of different ploidy levels were acclimatized using different media types while controlling humidity and preventing fungal contamination. There was no significant difference between Jiffy-7 pellet and loamy soil to produce higher numbers of acclimatized plantlets than soil mix of loamy soil: compost; 3: 1 (Table 2 and Fig. 2H). As the loamy soil can be locally sourced, it is more plentiful and cheaper than the Jiffy-7 pellet and, therefore, should be used as the main medium for acclimatization. Loamy soil provides adequate nutrients to sustain the growth of the plantlets apart from acclimatizing. The addition of compost into the loamy soil does not contribute to better growth or more successful acclimatization of the plantlets at the acclimatization stage. This study agrees with Mok & Ho (2019), where Jiffy-7 provides a high percentage of successful acclimatization. Jiffy-7 pellet is only made of compressed peat moss and coco fibers in the fine net that provides adequate aeration water. Mengesha, Ayenew & Tadesse (2013) obtained a similar result where Jiffy-7 pellet was found to improve not only acclimatization of in vitro plantlets of Ananas comosuss but also the growth during acclimatization.

The approach to humidity control in this study during acclimatization was similar to the method described by Ahmed et al. (2012). Newly deflasked plantlets of N. cadamba are susceptible to water stress and low humidity. The newly transplanted plantlets were placed in a clear, air-tight container to maintain high humidity. The container cover was removed slowly to expose the plantlets progressively to ex vitro conditions. Adapting the plantlets slowly to ex vitro conditions can prevent abiotic stress to the plantlets, which is the main cause of mortality. According to Chandra et al. (2010), high humidity in in vitro culture vessels can impair the stomatal apparatus function and retard the cuticle and epicuticular layer formation. This will lead to excessive transpiration when the plantlets are transferred to ex vitro environment with low humidity, high light intensity and wide temperature fluctuation. In nature, the N. cadamba trees thrive near to the river banks, which experience intermittent submerges in river water due to tidal phenomenon or rainfall (Mansur & Tuheteru, 2010). Therefore, regular watering is essential as the leaves tend to wilt during water deficiency. Plantlet loss can be caused by fungal infections found on either the roots, leaves or stems of the plantlets. The plantlets were misted with 5 gL−1 Benocide solution weekly using a mist sprayer to reduce fungal infections.

Morphological characteristics and SPAD value

This study observed that the octoploid and mixoploid plants of N. cadamba produced thicker leaf than the tetraploid plants (Table 3 and Figs. 3A–3C). This characteristic is in line with several other induced polyploid species, such as Anthurium andraeanum (Chen et al., 2011); Lobularia maritima (Huang et al., 2015); Lycium rhuthenicum (Rao et al., 2019); Manihot esculenta (Zhou, Zeng & Yan, 2017); Bacopa monnieri (Inthima & Sujipuli, 2019); Plumbago auriculata (Jiang et al., 2020) and Solanum lycopersicum (De Alencar, Azevedo & Latado, 2020). Detailed histological study on S. lycopersicum leaf blade revealed that thicker leaf in the induced polyploid was the result of greater thickness in the epidermis (adaxial and abaxial) and the parenchyma (spongy and palisade) layers (De Alencar, Azevedo & Latado, 2020). In the induced polyploid of Plumbago auriculata, the histological study showed that thicker spongy tissue contributed to thicker leaf (Jiang et al., 2020). These histological studies reveal the thicker leaf compositions but do not provide further information regarding the function of the thicker leaf in plants.

There is much empirical evidence showing that plants with thicker leaves have better adaptability to a drier environment. According to Afzal, Duiker & Watson (2017), plant species with thicker leaves can withstand greater drought and salinity. They also stated that leaf thickness assessment is a promising technique to predict water status in plants. In another study, under arid conditions, plants with thicker leaves can adapt by maintaining water potential when the water source is limited (Ogburn & Edwards, 2010). A thicker leaf is associated with the plant ability to inhabit and adapt to challenging environments like arid conditions and high irradiance (Li et al., 2014; Coneva & Chitwood, 2018). A plant can adapt to a harsh environment inflicted by climate change by modifying leaf morphological and anatomical traits (Dar et al., 2013; Soudzilovskaia et al., 2013; Tian et al., 2016; Ribeiro et al., 2016; Souza et al., 2018). Inferring from the evidence as stated, thicker leaf of N. cadamba plants acquired through polyploidization may similarly be more adaptable to a drier condition, such as in locations with lower rainfall. This improvement is essential to N. cadamba due to looming climate change exacerbated by global warming.

This study observed that the leaf midrib of the octoploid of N. cadamba is thicker than its progenitor (Table 3 and Figs. 3A–3C). This is in line with the finding of the polyploidization study of Salix viminalis by Dudits et al. (2016), where induced tetraploid plants were reported to have thicker midrib. This could increase water volume transported to the leaf blade where more water can be stored to prevent the plant from excessive water loss during drought. Furthermore, thicker leaf midrib provides a stronger leaf structure of N. cadamba that may prevent the leaf blade from wind damage.

Stomata morphology is commonly used to determine how polyploidization has modified the characteristics of the polyploid due to its ubiquity, homology across distantly related relatives and uniform shape (Doyle & Coate, 2019). This makes stomatal study reliable and repeatable when different leaf samples are used, apart from the relatively easier and faster (Xie et al., 2015; Eng & Ho, 2019). Polyploidization of plants always produces bigger stomata size and lesser stomata density (number of stomata per unit area) at the abaxial side of the leaf of different species such as Impatiens walleriana (Ghanbari et al., 2019); Dendrobium officinale (Pham et al., 2019); Lycium ruthernicum (Rao et al., 2019); Paphiopedilum villosum (Huy et al. 2019); and Plumbago auriculata (Jiang et al., 2020). These characteristics can also be observed in N. cadamba in this study, where octoploid and mixoploid plants pose bigger stomata in terms of mean length and mean width and lower mean stomata density compared to their tetraploid progenitor (Table 3 and Figs. 3D–3F). A negative correlation between stomata size and stomata density is common in the leaves (Camargo & Marenco, 2011; Doheny-Adams et al., 2012; Bertolino, Caine & Gray, 2019). This contrasting characteristic is called plastic developmental responses to evolutionary adaptation and environmental changes (De Boer et al. 2016; Dittberner et al., 2018).

There is no structural difference of stomata among the tetraploid, mixoploid and octoploid plants of N. cadamba (Figs. 3G–3I). The N. cadamba stomata belong to the paracytic type, where each guard cell is accompanied by a subsidiary cell to its long axis (Willmer & Fricker, 1996). This stomata type is typical to members of the Rubiaceae family. Most water obtained in plants is transpired through stomatal apertures before being utilized for plant growth. The ability to conserve water and use it efficiently will improve plant growth. Stomata characteristics, such as size and density, play an essential role in determining water-use-efficiency in the crop, thus making these characteristics essential to crop improvement target (Bertolino, Caine & Gray, 2019). Under a controlled atmosphere, Arabidopsis thaliana mutant plants with lower stomatal density are better adapted to the higher concentration of atmospheric CO2 and water-scarce environment (Doheny-Adams et al., 2012). This finding suggested that genetically modified plants with lower stomatal density can be better adapted to predicted future climate change where elevated atmospheric CO2 andmore arid land become more prevalent. Recent research has also shown that genetically engineered crops with reduced stomatal density may contribute to better water usage efficiency and drought tolerance without productivity decline (Bertolino, Caine & Gray, 2019). Given much evidence, induced octoploid and mixoploid of N. cadamba with reduced stomata density and increased stomata size are likely to attain a greater advantage in utilizing water resources in the field.

Polyploidization of N. cadamba caused significantly reduced trichome density of mixoploid and octoploid plants than their tetraploid progenitor plants. However, the trichome size of mixoploid and octoploid plants were relatively bigger than tetraploid plants (Table 3, Figs. 3J–3L). This finding is in line with Zhang & Gao (2020), where trichomes density on leaf sheath of Dendrobium cariniferum has reduced in the induced tetraploid plant compared with the diploid progenitor plant. The trichomes play a vital role in protecting the plant from biological and environmental hazards, namely herbivores, pathogens, excessive ultraviolet irradiation and over transpiration (Xiao et al., 2017). The characterization and classification of trichomes are challenging due to the immense diversity of trichomes in morphology, origin, size, location, surface microstructure, secretory organ, function and timing of activity. However, the main characteristic is glandular or non-glandular (Werker, 2000). The trichomes of the tetraploid, mixoploid and octoploid of N. cadamba are categorized as non-glandular and unicell trichomes. The shapes and attributes of the trichomes have been described in detail by Payne (1978) and the trichomes of N. cadamba polyploid plants are termed as ornithorhynchous, which resembles a bird’s bill (Figs. 3J–3L).

Stem cross-section of N. cadamba showed that the pith of either the tetraploid or mixoploid was significantly bigger than that of the octoploid (Table 3 and Fig. 3M). However, there is no significant difference for wood and bark layers of tetraploid, mixoploid and octoploid plants. The smaller pith in the octoploid is an improvement to the stem of N. cadamba, which may be of importance to wood production. The pith in N. cadamba will eventually dry up as the tree ages or after harvesting. The stem cross-section study is destructive as the plant has to be sacrificed to reveal the stem composition. Stem cross-section has rarely been studied in the polyploidisation of plants. The stem cross-section study is important to those trees that are used for timber production and where the quality and quality of the timber are desired. In the Salix viminalis polyploidization study, the induced tetraploid plants have a significantly greater composition of bark and wood layers than the tetraploid plants (Dudits et al., 2016).

The SPAD value of induced polyploid plants is often studied to determine their chlorophyll content (Wang et al., 2017; Rao et al., 2019). Our study is in line with both studies showing induced polyploid plants of N. cadamba gain a higher mean SPAD value than their progenitor (Table 3). SPAD value correlates positively to the chlorophyll content and greenness of the leaf measured (Limantara et al., 2015). Increased chlorophyll content in induced polyploids may show increased photosynthesis rate (Wang et al., 2017; Rao et al., 2019). From these findings, the mixoploid and octoploid leaves are predicted to have higher photosynthetic capacity due to their higher SPAD value or higher content of chlorophyll. The improvement in SPAD value in polyploid of N. cadamba signifies the importance of polyploidization as a plant breeding tool. The SPAD value is not the sole factor that results in greater growth or yield of a crop. In wheat, the relationship between SPAD value and grain productivities was studied (Monostori et al., 2016). They found that other factors also affect the grain productivity, such as soil nitrogen content, cultivar and environment. According to Kandel (2020), SPAD value and yield are not often collinearly related as it may result from genetic variation, differential absorption strength in nitrogen, chlorophyll biosynthesis and chlorophyll degradation.

Conclusions

To the best of our knowledge, this is the first report of polyploidization in N. cadamba. Our study lays the foundation for producing octoploid and mixoploid plants of N. cadamba using an anti-mitotic agent—colchicine. Both colchicine-induced polyploid plants are growing slower than tetraploids under in vitro conditions. However, the leaves of octoploid plants have thicker leaf blades, thicker midrib, lower stomata density and bigger stomata size. This indicates that octoploid plants could respond better to environmental changes. These novel polyploid plants could be beneficial for future genetic improvement of N. cadamba; for instance, octoploid could backcross with tetraploid to produce a novel hexaploid plant (2n = 6x = 66) or interploid. Hence, the in vitro colchicine-induced mutation could be a valuable breeding strategy of N. cadamba to produce improved clones for planted forest development.

Supplemental Information

Supplemental Information 1 Raw Data

Click here for additional data file.

Supplemental Information 2 Supplementary Tables

Click here for additional data file.

The authors would like to acknowledge WTK for providing the plant materials for this research project.

Additional Information and Declarations

Competing Interests

Author Contributions

Data Availability

Kwong Hung Ling is the Chairman of the STA Forest Plantation Committee Category.

Wee Hiang Eng and Wei Seng Ho conceived and designed the experiments, performed the experiments, analyzed the data, prepared figures and/or tables, authored or reviewed drafts of the paper, and approved the final draft.

Kwong Hung Ling analyzed the data, authored or reviewed drafts of the paper, and approved the final draft.

The following information was supplied regarding data availability:

The ANOVA tables are available in the Supplementary File.

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
