# Peer review of "In vitro induction and identification of polyploid Neolamarckia cadamba plants by colchicine treatment"

_PeerJ, doi:10.7717/peerj.12399_

## Round 0.1 · original submission · Major Revisions

Your manuscript has now been reviewed by two referees. Their comments are included below. Both reviewers and I agree that this study is original and within the aims and scope of PeerJ. However, I do not think that the manuscript is quite ready for publication in its present form and therefore suggest reconsideration of the manuscript after revision. Both reviewers raise good points and I would like to ask the authors to address all of these carefully. Specifically, this paper needs substantial revision in the Introduction and Discussion. Discussion should be shortened (authors should avoid repeating the results and speculative sentences). The number of figures seems excessive so I recommend combining them (see referee #1 comment). Units should be checked in the Result section (for stomata measures, please use micrometers). After you have made all the changes requested, please upload a point-by-point response to the referees' reports.

Reviewer 1 ·

Basic reporting

The authors reported colchicine-induction of polyploids of Neolamarckia cadamba here. In vitro regeneration, in vitro and on soil growth, and morphological characters were also compared between the original tetraploids, induced octaploids and mixaploids. The data are clear and described in detail. Discussion part is a little bit too wordy, and could be reduced in length, but overall contents are well prepared. Specific comments are as follows:
Introduction.
1) In line 50, the authors are better to mention to the synonym Anthocephalys cadamba Roxb,, that will be referred in Line 362. Unless readers cannot understand the part around the Line 362 of Discussion part.
2) It is strongly recommended to mention to the original ploidy level (tetraploid, 2n = 4X) first in the Introduction. Unless, as myself, readers would misunderstand that the authors obtained tetraploids and octaploids from diploid original until the Discussion part.
3) Line 63, colchicine is NOT "anti-microtubule" drug, but inhibits polymerization of the tubuline.
4) LIne 99-100, "Machine" is not appropriate here. Just "FCM" is better.
5) Line 100-102, while FCM is high-throughput to examine the ploidy level, it is hard to detect aneuploidy. Therefore combination of FCM and karyotype examination is the best way. The authors have to mention to the above here. In many past studies, colchicine-induced doubling of the ploidy often cause aneuploidy, too. Did not authors find such case here?
6) Line 111 "there is no published..." is better to be said as "this is the first report..."
Materials and methods
7) Line 126, how much was the sucrose concentration ?
8) Line 142 "5-days-old": from when? 5 days after the transplantation?
9) ibid. "between 9 00 to 10 00 am": this is not informative and no need.
10) Line 213 "SPAD": at the first appearance this type of abbreviations should be written in a full spell.
Results
11) Line 287-288, such as "94.44%". These figures are too fine. Please re-examine the significant digits. This is also the case for Line 298-300 as "0.247 mm" (or is this 247.0 micrometer?). Please check the significant digits also for the tables 2 and 3.
Discussion
12) Something was lost between line 368-369.
13) In overall meaning, the Discussion part is too wordy. For example the paragraph from line 394-408 could be omitted, because this is too general matter.
14) Line 446, "PGR": at the first appearance this type of abbreviations should be written in a full spell.
15) Line 576 "Arabidopsis" should be "Arabidopsis thaliana". For your reference, Tsukaya (2008) doi:10.1371/journal.pbio.0060174 referred to a "high-ploidy syndrome" in the octaploid of Arabidopsis thaliana.
References
16) Order should be re-examined. For example, Sharma and Sen should be after the Sen et al.
Figures
17) Some could be combined, for example Figures 3 with 4 and 5, or 9 with 10.

Experimental design

Well designed. Minor points were raised above.

Validity of the findings

Well confirmed.

Additional comments

Detailed comments are available above "1 Basic reporting".

·

Basic reporting

This manuscript nicely documents chromosome doubling and associated morphological changes in a Malaysian timber species. The methods and results are well-presented, the figures and tables are appropriate. Some critical information is missing from the introduction – detailed below. The discussion includes an abundance of general background information and is much too long for the results described here – sections that can be deleted are listed below. Some potential outcomes of this study are presented as facts – these need to be rephrased. For example, the morphological changes documented here may contribute to increased photosynthetic capacity, but this remains to be evaluated. Likewise, the octoploid may be valuable in cultivation, but this remains to be determined.

Experimental design

no comment

Validity of the findings

see section 1.

Additional comments

Line 28: It is premature to state that polyploidy “has become a valuable breeding strategy” since you are reporting it for the first time here.
Line 28: Please provide the family and describe that Neolamarckia cadamba is a tetraploid with 2n=44. Cite the previous chromosome counts.
Line 34: Define mixoploidy as mixed tetraploid and octoploid cells, when first used in abstract and text.
Line 39: “grew” not “reported growing”
Line 42: Morphology is not “improved”. It changed and resulted in traits that are predicted to increase phyotosynthetic capacity”
Line 50: Please provide the family and state that Neolamarckia cadamba is a tetraploid with 2n=44.
Line 54: that is easy to work…
Line 69: delete “still”
Lines 83-84: sentence can be deleted
Line 97: define mixoploid
Line 102: straightforward
Line 111: change “there is no published report on the polyploidization study of N. cadamba” to “there is no previous report of polyploidization in N. cadamba”
Line 260: replace “caused lower renderability” with “resulted in higher mortality”
Line 312 “possess” not pose
Lines 341-346: The description of explant mortality needs to be moved to the results.
Line 362: Please add that Anthocephalus cadamba is the same as Neolamarckia cadamba. Better yet add this to the introduction, and then you say “N. cadamba (reported as A. cadamba)” here.
Lines 369-464. Most of this is general background information or repeats the results, and can be deleted. Please keep the citations of previous chromosome counts, but move these to the introduction.
Line 547: replace “The stomatal study” with “stomata morphology”
Line 559-599. General background information on stomata can be deleted.
Line 632: replace “have” with “are predicted to have”
Line 646: delete “were more superior compared with the tetraploid; the former”
Line 652-654: very speculative and best deleted

---

## Round 0.2 · accepted · Accept

Both reviewers found that the authors adequately revised the manuscript and it is ready for publication. The two minor suggestions of Reviewer #1 should be addressed in the proof correction stage if they are not corrected earlier.

Reviewer 1 ·

Basic reporting

I found that the authors adequately revised their text and figures, except for a few points.

1) Line 125. The authors added "2% sucrose". Here they must precisely write it as "2% (w/v) sucrose", perhaps.

2) Literature. Li, Q. et al. must be before the Li, S. et al., I think.

Figure combinations are now quite improved.

Experimental design

Good.

Validity of the findings

Enough good.

Additional comments

As mentioned above in the basic reporting, a few minor points should be fixed.

·

Basic reporting

The authors have appropriately addressed all reviewer comments. The revised figures are much improved.

Experimental design

blank

Validity of the findings

blank

Additional comments

none